# Research on Multiple Spectral Ranges with Deep Learning for SpO_2_ Measurement

**DOI:** 10.3390/s22010328

**Published:** 2022-01-02

**Authors:** Chih-Hsiung Shen, Wei-Lun Chen, Jung-Jie Wu

**Affiliations:** Department of Mechatronics Engineering, National Changhua University of Education, Changhua 500, Taiwan; hilbert@gm.ncue.edu.tw (C.-H.S.); d9951003@gm.ncue.edu.tw (J.-J.W.)

**Keywords:** SpO_2_, convolution neural network, multi-wavelength

## Abstract

Oxyhemoglobin saturation by pulse oximetry (SpO_2_) has always played an important role in the diagnosis of symptoms. Considering that the traditional SpO_2_ measurement has a certain error due to the number of wavelengths and the algorithm and the wider application of machine learning and spectrum combination, we propose to use 12-wavelength spectral absorption measurement to improve the accuracy of SpO_2_ measurement. To investigate the multiple spectral regions for deep learning for SpO_2_ measurement, three datasets for training and verification were built, which were constructed over the spectra of first region, second region, and full region and their sub-regions, respectively. For each region under the procedures of optimization of our model, a thorough of investigation of hyperparameters is proceeded. Additionally, data augmentation is preformed to expand dataset with added noise randomly, increasing the diversity of data and improving the generalization of the neural network. After that, the established dataset is input to a one dimensional convolution neural network (1D-CNN) to obtain a measurement model of SpO_2_. In order to enhance the model accuracy, GridSearchCV and Bayesian optimization are applied to optimize the hyperparameters. The optimal accuracies of proposed model optimized by GridSearchCV and Bayesian Optimization is 89.3% and 99.4%, respectively, and trained with the dataset at the spectral region of six wavelengths including 650 nm, 680 nm, 730 nm, 760 nm, 810 nm, 860 nm. The total relative error of the best model is only 0.46%, optimized by Bayesian optimization. Although the spectral measurement with more features can improve the resolution ability of the neural network, the results reveal that the training with the dataset of the shorter six wavelength is redundant. This analysis shows that it is very important to construct an effective 1D-CNN model area for spectral measurement using the appropriate spectral ranges and number of wavelengths. It shows that our proposed 1D-CNN model gives a new and feasible approach to measure SpO_2_ based on multi-wavelength.

## 1. Introduction

The oxyhemoglobin saturation by pulse oximetry (SpO_2_) is an important health indicator. In pulse oximetry, the oxyhemoglobin saturation of a healthy person is usually 95–100%, and a reading below 90% is considered a dangerously low value. Since shortness of breath is the main symptom of severe COVID-19, pulse oximetry is used as an early warning sign to track their blood oxygen levels in case patients with supplemental oxygen in the hospital. SpO_2_ monitoring helps doctors understand whether the patient’s vital signs are stable and make appropriate treatment immediately.

At present, the measurement of SpO_2_ with photoelectric technology uses red light near 660 nm and near-infrared light near 940 nm as the light source [1,2], with PPG (Photoplethysmography) to read the light intensity and convert the signal into sequence data for further analysis. PPG has developed into two types of measurement principle, namely a transmission-type and reflection-type. The reflection system is common in wearable devices, which is prone to influence by the measurement position of the subject or the thickness of the skin tissue and affects the intensity of light reflection and the accuracy of the measurement. The most common transmission system is implemented with a finger clip, which is used to fix and measure thinner skin tissue. The light intensity received by measuring thin skin tissue is about 40 to 60 decibels higher than the light received by the reflection system.

The most primitive method of tracing oxygen saturation requires collecting a blood sample first and then perform electrochemical analysis. This method has a complicated process and is unable to achieve continuous measurement. However, with the development of clinical medicine, non-invasive oxygen saturation measurement is now generally used. Wearing a finger clip with photoelectric sensor can realize continuous SpO_2_ measurement. This method was proposed by Robert Andrews Millikan. A red light with a wavelength of 660 nm and near-infrared light with a wavelength of 940 nm as measuring light sources to monitor the ratio of oxyhemoglobin and deoxyhemoglobin in the arteries and calculate SpO_2_.

In terms of measurement resolution and accuracy, the number of features that can be extracted from dual-wavelength data are limited. In the previous research works, Jain Anant has revealed some drawbacks for dual-wavelength pulse oximeters in 2021. It shows that the SpO_2_ value calculated by the traditional pulse oximeter by weighted moving average. This method is mainly susceptible to motion artifacts, background light, and low perfusion state errors. Spectral analysis has been identified as a good way to improve calculations [3]. Besides, the previous research works by T. Aoyagi and M. Fuse have shown that three factors affect pulse oximetry: the optics, the tissue, and the venous blood. They conducted a physical theoretical formula of pulse oximetry, based on their research, the three-wavelength method eliminated the effect of tissue and improved the accuracy of SpO_2_. The five-wavelength method eliminated the effect of venous blood and improved motion artifact elimination. Their research indicated that multi-wavelength improves most of the defect of traditional method using only two bands [4] for solving problems in pulse oximetry such as accuracy, motion artifact, low-pulse amplitude, response delay, and errors, and provided an approach the possibility of multi-wavelength pulse oximetry. Since the signal to noise ratio is proportional to the square root of the number of wavelengths considered, the more wavelengths measured, the more robust the oximetry measurement can be [5]. It is believed that increasing the number of transmission wavelengths may improve the accuracy of the measurement, and also can monitor other carbon monoxide in the blood or wastes of liver and kidney metabolism at the same time [5,6]. Therefore, this study uses multi-wavelength to enhance the accuracy of measurement.

The traditional algorithm always needs calibration. However, it depends on the volunteers selected. It is related to human skin conditions and many other factors, so it is difficult to obtain accurate calibration coefficients. That is why many researchers have to propose new methods to improve the measurement [7,8,9]. In the traditional algorithm, there is a constant Q used to express the light absorption ratio of two wavelengths in the transmission equation. This constant adjusts the error in the actual measurement of arterial blood due to continuous pulsation. λ_1_ and λ_2_ represent the wavelength of the light source. DC_λ1_ and DC_λ2_ are the sum of the light intensity of λ_1_ and λ_2_ absorbed by the non-pulsatile part of the artery and the venous blood tissue, and AC_λ1_ and AC_λ2_ are the light intensity of λ_1_ and λ_2_ absorbed by the pulsatile arterial blood. Through the derivation in Equation (1), Equation (2) is an approximate formula, it is inevitable causing errors in the final result. Compared with the above algorithm, deep learning learns multi-wavelength spectral absorption curves through its hidden layer architecture gradually without the need of complicated algorithm. In this study, the neural network with deep learning architecture will identify SpO_2_ concentration of numerical labels via training beyond the traditional models. Each neuron or node of output layer corresponding to the SpO_2_ concentration in the network represents a label of the output.

We present here the novel use of deep learning architecture with one-dimensional convolutional neural networks (1D-CNN) to overcome the aforementioned shortcomings in current retinal oximetry model for directly quantifying SaO_2_ and SpO_2_ concentration concurrently in a reproducible, robust way. Deep learning architecture has been finding more and more applications in the biomedical field both in image recognition and spectral identification. For spectroscopy, deep learning architecture with 1D-CNN have incredible advantage over statistical regression techniques, owed mainly to their ability to learn and weigh the importance of different spectral regions automatically. Furthermore, they learn this weighting of spectral characteristics with no prior knowledge of the constituent’s absorption spectra. This is extremely interesting for us to develop SpO_2_ spectroscopy with multi-wavelength spectral absorption based on the deep learning architecture. Furthermore, Equation (2) is applicable to dual-wavelength measurement, which use the ratio of Q to further calculate the SpO_2_. For the calculation of multi-wavelength, there does not exist a general formula to give the SPO_2_ from multiple inputs from the reading of spectral meter. Especially for more thorough analysis of multi-wavelength measurement, the number of inputs need not be fixed and several different ranges of spectral detection are required and analyzed in the work. Hence, a 1D-CNN is used in this study as a method to achieve the multi-wavelength measurement which isn’t fixed on the number of wavelengths. In this study, the model input is transmission spectra, and the model output is ten SpO_2_ values.
(1)$$Q=\frac{\ln\frac{{\mathrm{DC}}_{\lambda 1\;}-{\;\mathrm{AC}}_{\lambda 1}}{{\mathrm{DC}}_{\lambda 1}}}{\ln\frac{{\mathrm{DC}}_{\lambda 2\;}-{\;\mathrm{AC}}_{\lambda 2}}{{\mathrm{DC}}_{\lambda 2}}}=\frac{\ln\left(1\;-\frac{{\mathrm{AC}}_{\lambda 1}}{{\mathrm{DC}}_{\lambda 1}}\right)}{\ln\left(1\;-\frac{{\mathrm{AC}}_{\lambda 2}}{{\mathrm{DC}}_{\lambda 2}}\right)}\;$$
from $\frac{{\mathrm{AC}}_{\lambda 1}}{{\mathrm{DC}}_{\lambda 1}},\;\frac{{\mathrm{AC}}_{\lambda 2}}{{\mathrm{DC}}_{\lambda 2}}\;\ll \;1$
(2)$$Q=\frac{\frac{{\mathrm{AC}}_{\lambda 1}}{{\mathrm{DC}}_{\lambda 1}}}{\frac{{\mathrm{AC}}_{\lambda 2}}{{\mathrm{DC}}_{\lambda 2}}}$$

Medical data can generally be divided into two types: images and numerical values. At present, most of the applications in two-dimensional CNN are applied to identify image data. However, spectral data is part of the sequence value, and it is not directly related to two-dimensional CNN. Therefore, 1D-CNN will be a powerful tool for us to achieve our goals.

1D-CNN is becoming more and more widely used in medical treatment currently. For example, in 2020, Ramis İleri and other researchers detect EOG (Electrooculogram) signals around the eyeballs and use 1D-CNN to determine whether the subject has dyslexia. Finally, the accuracy of their experimental results was 73.6128 ± 2.8155% [10]. Besides, 1D-CNN is widely used to achieve detection and diagnosis of symptoms [11,12,13,14]. It is obvious that 1D-CNN has high potential in judging numerical data.

Before we use 1D-CNN, our previous study has used deep neural networks (DNN) to train a predictive model to measure SpO_2_ [15]. In the study, 12-channel spectra are used to train the DNN model constructed by three hidden layers with 200 neural, and finally a single neuron is used as the output. It is different from 1D-CNN prediction that output the probability of each category. 1D-CNN’s operation mode can be regarded as a scalar inner product of many different filters and spectral data. Each filter is composed of patches with different weights. These filters that move on the data are also called kernel maps. It moves in a single fixed direction on the sequence data. In this way, the neural network will automatically extract the important features of each data as the identification basis. Equation (3) shows the mathematical expression of the 1D-CNN operation, where H_k_ is the length of the convolution kernel, U_k_ is the length of the sequence data, and k is the number of steps required to scan a single data. Besides, *h_k-i_* is the *i*_th_ row of convolution kernel and *u_i_* is the *i*_th_ row of sequence data. In addition, 1D-CNN has a fixed width of convolution kernel, it is unnecessary to specified the width separately. Therefore, the input shape of the 1D-CNN is a 3D tensor, which is batch size, steps, and input dimension in order.
(3)$$Y\left(k\right){=H}_{k}U_{k}=\overset{N}{\underset{i=0}{\sum }}h_{k-i}u_{i}$$

For a successful neural network model, it is necessary to have appropriate and high-quality training data. In 1999, Moritz Friebel and other researchers analyzed blood from 400 nm to 2500 nm [16]. They showed the absorption coefficient graph of Hb and HbO_2_ in Figure 1. From the figure, the absorption coefficients from 600 nm to 1100 nm are highly distinguishable. Therefore, this study chooses the long wavelength range of visible light to infrared light as the training data of neural network.

In this study, multi-wavelength sensor is chosen to meet the requirements of the ideal measurement wavelengths. Due to breakthroughs in spectral measurement technology, this sensor has a lower cost and a smaller size. In 2019, J.-S. Botero-Valencia and other researchers used these sensors, and a neural network to convert the multi-channel light intensity that was originally discrete in the spectra into a continuous curve [17]. It also shows that the SpO_2_ value calculated by the traditional dual-wavelength pulse oximeter is mainly susceptible to motion artifacts, background light, and low perfusion state errors. Spectral analysis has been identified as a good way to improve calculations. Several researches show that multi-wavelength improves most of the defect of traditional oximeter. We propose to use 12-wavelength spectral absorption measurement to improve the accuracy of SpO_2_ measurement, and build different datasets according to the spectral characteristics. Besides, dataset is added with noise randomly, increasing the diversity of data and improving the generalization of the neural network. The final result shows that a spectrometer constructed through a neural network has a resolution of up to 5 nm and a maximum error of less than 2%, which shows the necessity of multi-wavelength spectral measurement for SpO_2_.

## 2. Materials and Methods

### 2.1. Materials

In this study, the setup of measurement includes three parts, including light source, light sensor and the architecture of light sensing, and signal processing.

#### 2.1.1. Light Source

Since the light transmission-type of SpO_2_ measurement is used in this study, a measuring light source that contains both visible light and infrared light having a stable light intensity is indispensable. On the other hand, a xenon lamp is used with high luminous efficiency, long life, and the emission spectral range (400~1100 nm) completely covers the range required for research. In order to stabilize the light source and temperature, a 6.4 V constant voltage power supply is adapted to replace the original battery for UltraFire 9P xenon flashlight to maintain a better power stabilization.

#### 2.1.2. Light Sensor

The multi-wavelength measurement including visible light and infrared light ranges was defined for our measurement and 1D-CNN neural network architecture. Two six-channel sensors, AS7262 and AS7263, are used, which cover visible light and infrared light ranges and conduct measurement of transmission light. AS7262 captures six wavelengths of 450 nm, 500 nm, 550 nm, 570 nm, 600 nm, and 650 nm, and its full width at half maximum (FWHM) is 40 nm. AS7263 captures the remaining six wavelengths, which are 610 nm, 680 nm, 730 nm, 760 nm, 810 nm, and 860 nm, and its full width at half maximum (FWHM) is 20 nm. Among these 12-channel, the NIR (near infrared region) area sensed by AS7263, will become the focus of our discussion.

Two embedded modules are used to acquire data from two sensors and read the light intensity value through I^2^C communication protocol. The data of these 12-channel are read by the two embedded modules and transmitted to the edge computing system for SpO_2_ calculations.

Using a finger clip to fix the finger position can improve the stability of the measurement data during the measurement process. The 3D printed finger clip designed according to human’s finger is used in this study, as shown in Figure 2. The shield of ambient light is used on the end of the finger to ensure the sensor to receive the light transmitted by fingers. There is a rectangular window on the top for light projection, and a pupil at the bottom for penetrated light transmission.

In order to make the equipment more stable during the measurement process, a 3D printed light sensing case beneath the finger clip device is designed and fabricated, as shown in Figure 3.

#### 2.1.3. Architecture of Light Sensing and Signal Processing

Raspberry Pi 4 Model B is selected as the edge computing module for result computing and historical record maintenance. The spectral signal from the embedded modules can be sorted out, and input into the pre-trained neural network model for measurement, and displayed on the screen.

### 2.2. Measurement

Figure 4 is the complete measurement configuration of the equipment. The light source emitted by the xenon lamp and focusing on the finger clip through the convex lens. When the subject puts the index finger of the left hand into the finger clip, the light source will project on the skin close to the nail, and the attenuated light penetrating the finger will pass through the beam splitter and project on the AS7262 and AS7263 light sensors. The measurement accuracy of this system is complicated and mainly relies on several major devices which limit the accuracy and resolution of measurement. Firstly, the two six-channel sensors, AS7262 and AS7263 are used with 16 bits A/D converter which gives ±1 LSB Max (±0.0015% of Full Scale) with no missing codes. The measurement covers the visible light and infrared light ranges with peak sensitivities at six wavelengths, each with 40 nm/ 20 nm full width at half maximum (FWHM) for AS7262 and AS7263, respectively. Table 1 shows some important features of the AS7262/AS7263 sensors.

The illuminated intensity on the fingers from Xenon light source is calibrated and normalized to eliminate the temperature drift of intensity. The SpO_2_ data are acquired by a standard meter, Rossmax SB100 with 1% resolution which is used to calibrate and label the output of training dataset from our proposed model. The accuracy of Rossmax SB100 is within ±2% over the range of SpO_2_, 70~99%. Moreover, the signal obtained by the sensor is transmitted to the embedded modules through the DuPont cable, and the reading time interval of the light sensor is 0.7 s due to the 0.6 s for integration time of light sensors and 0.1 s for the computation of edge computing module. Finally, the data is transmitted to the neural network model, which was trained in Raspberry Pi to make continuous measurement of SpO_2_. The results will be displayed on the screen with the designed human-machine interface and recorded in chronological order at the same time automatically.

## 3. Architecture of 1D-CNN and Hyperparameters Optimization

The experiment flow chart is shown in Figure 5. The experimental process is divided into three stages. There are data establishment, 1D-CNN model configuration and training, and equipment execution and verification. The focus of this study is the second stage. Besides, there are three issues in this experiment that need to be analysis, training with different spectral channels, random noise addition, and hyperparameters optimization, respectively.

### 3.1. Data Establishment

#### 3.1.1. Data Augmentation

When the experiment and training of neural networks are performed, two issues need to be considered, namely over-fitting and under-fitting. It is a challenge that trained model can complete the SpO_2_ measurement data under all conditions, especially when the SpO_2_ measurement is performed in the actual subjects, the actual experimental dataset is not easy to get acquisition with all conditions thoroughly. When a neural network is trained with a small dataset, this will cause the network remember the training dataset instead of learning the general characteristics of our experimental SpO_2_ measurement data. For this reason, the model performs well on the training dataset but does not perform well on the test dataset. When a small dataset provides a bad description of our problem, it may lead to a problem that is difficult to learn. Obtaining more data is a very expensive and difficult task when SpO_2_ is below 95%. At this point, two techniques including noise addition and normalization are adopted to obtain better model performance. In this research, the noise on neural networks will be applied and be analyzed in detail. This technique not only reduces overfitting, but also optimizes our model faster and the overall performance is improved noticeably.

#### 3.1.2. Preprocessing and Construction of Dataset

When we collect neural network training data, traditional measurements require blood tests, and a large amount of continuous data cannot be obtained. Although spectral analysis has been identified as a good way beyond the dual-wavelength pulse oximeters, in order to achieve calibration, the SpO_2_ data are acquired by a standard meter, Rossmax SB100 under a severe and fixed conditions to avoid the drawbacks of dual-wavelength pulse oximeters.

In order to obtain reliable and valid data, the SpO_2_ of right hand is measured by a standard meter, Rossmax SB100, and the SpO_2_ of left hand is measured by the constructed measuring device to simulate hypoxia by holding breath to obtain the spectra and SpO_2_ at the same time. 215 sets of measured light intensity of each wavelength and the corresponding time is recorded, which is helpful for labeling the SpO_2_ of standard meter. Finally, the multi-wavelength spectra of SpO_2_ from 81% to 99% is completed.

High-quality data must have completeness and reliability. Therefore, the multi-wavelength spectral data collected by the measuring device is normalized between 0 and 1 by Equation (4), which is denoted as I_normalized_, as shown in Figure 6. Furthermore, I_max_ is the maximum of the spectral data, and I_min_ is the minimum of the spectral data. Normalized data have two major benefits. First, they can effectively reduce the number of iterations required by the gradient descent method. Secondly, they makes the data in different dimensions comparable without concerning the accuracy influence of a large number in certain dimensions.

The data with the least noise interference in each concentration is selected as the effective data, which is denoted as I_eff_. The SpO_2_ dataset is created in an interval of two with SpO_2_ 99%, 97%, 95%, 93%, 91%, 89%, 87%, 85%, 83%, and 81% separately to maintain excellent resolution for neural network measurement, as shown in Figure 6, and put it into the written program for preprocessing. Random noise with five conditions as 0%, 1%, 2%, 5%, and 10% is applied uniformly to expand the number of data and increase the generalization ability of the model. Hence, the neural network can be more accurate when facing different subjects.
(4)$$I_{\mathrm{normalized}}=\frac{I_{\mathrm{eff}}{\;-\;I}_{\min}}{I_{\max}{\;-\;I}_{\min}}\;$$

In the data configuration stage, the spectra of each SpO_2_ concentration is expanded to 1000 rows through the random noise to match the input dimension of the 1D-CNN. Each dataset will have 1000 rows of SpO_2_ spectral data, satisfying the input dimension of the neural network. Among the measurement in Figure 6, it is worth to noticed that the normalized spectra curve at the spectral region (450 nm, 500 nm, 550 nm, 570 nm, 600 nm, and 610 nm) denoted as first region is difficult to be distinguished since the spectral absorbance of oxyhemoglobin is relatively indistinguishable in this spectral range [13]. On the other hand, the normalized spectral curves at the second region (650 nm, 680 nm, 730 nm, 760 nm, 810 nm, and 860 nm) are easier to be distinguished since the spectral absorbance of oxyhemoglobin is relatively indistinguishable in this spectral range [13]. Hence, three datasets for training and three datasets for verification are built, which are constructed over the spectra of the first region, second region, and full region, respectively.

### 3.2. 1D-CNN Model Configuration and Training

In this study, the model configures with two convolutional layers and a pooling layer twice, followed by hidden layer and an output layer, shown in Figure 7. The convolutional layers and the hidden layers both use Relu as the activation function due to its high performance on convergence [18,19]. The output layer uses Softmax and cooperates with categorical cross-entropy as the loss function due to the excellent performance of Softmax when solving multi-category problem [20,21].

In terms of output, since the original output result of the CNN is the weight of each category, it is different from our ideal output. Therefore, the final result is the sum of multiplying the SpO_2_ labels and their weights of each category, shown in Equation (5), where *L_i_* is the *i*_th_ SpO_2_ label and *W_i_* is the *i*_th_ corresponding weight.
(5)$${\mathrm{SpO}}_{2}=\overset{10}{\underset{i=1}{\sum }}{\;L}_{i}W_{i}$$

### 3.3. Configuration of Optomechanical Sensing System

After completing the neural network training, the SpO_2_ measurement needs to be implemented by putting the model on our configured hardware device. Besides, there are three tasks for us to complete the measurement equipment. At the beginning, designing program and human-machine interface and insert into the edge computing module with the trained model is the most primary task in this stage. Secondly, although a light shield has applied on the sensor end, it cannot completely prevent the ambience light. Therefore, it is significantly to filter the noise out. Lastly, since the input shape required by 1D-CNN is sequential data, each input data must be expanded into the length set by the neural network model.

## 4. Analysis of Multiple Spectral Ranges

### 4.1. Analysis of Hyperparameters

To investigate the effectiveness of spectral regions for deep learning for SpO_2_ measurement, three datasets for training and verification are built, which are constructed over the spectra of the first region, second region, and full region and their sub-regions, respectively. For each region under the procedures of optimization of our model, a thorough of investigation of hyperparameters is proceeded firstly. Therefore, the estimated sensitive hyperparameters are analyzed by GridSearchCV which will not only search for the best parameters of the model, but also automatically cross-validate all training sets and retrain the model [22]. The first parameter analyzed is the dropout ratio which shows a sensitive affection of accuracy in our model. The appropriate dropout ratio can effectively prevent neural network from overfitting. Figure 8a shows the impact of different dropout ratios. On the accuracy of the neural network and the remaining hyperparameters are fixed and chosen during the grid searching of dropout parameter. After the searching, the neural network has the highest accuracy when dropout ratio is 0.35 which is marked with a shaded circle. The second relatively sensitive parameter is learning rate. As shown in Figure 8b, the results show it has the highest accuracy rate when the learning rate is 9 × 10^−4^, marked with a shaded circle. In this searching of parameters, if the learning rate is too high, the loss value will oscillate after converging to a certain value, and it will not converge to the global best solution. On the contrary, if the learning rate is too low, the convergence speed will be slowed, and a high-precision model may not be obtained due to trapped in the local optimum.

### 4.2. Optimization Procedures

In order to improve model performance and obtain high accuracy, it is necessary to optimize hyperparameters. GridSearchCV and Bayesian optimization (BO) are used to adjust the hyperparameters. GridSearchCV will search all of the parameters being assigned completely. BO will refer to the past results and constantly updates the probability model to focus the hyperparameters that may be the best solution, which will effectively increase the rate of searching [23]. Before deciding on the final searching scope, GridSearchCV and BO search coarsely, and a smaller searching range is arranged according the previous optimization results at the end. The searching conditions of hyperparameters with GridSearchCV and BO are shown in Table 2 and Table 3, respectively.

The automatic adjustment of hyperparameters can effectively acquire the optimal parameters of CNN and also improve the time cost of manual adjustment. At the beginning of optimization procedures, we compare the accuracy of the model, where hyperparameters generated by GridSearchCV and BO are shown at Table 4 and Table 5. The tables show that although most the model obtained by GridSearchCV has good accuracy, the performance of model using BO shows better results, which reveals that the model of this study can be optimized more effectively using BO.

#### 4.2.1. Investigation of the Validity of Spectral Regions

It is important to investigate the issues of validity of 1D-CNN Model to the spectral measurement and the accuracy and validation of spectral regions for different models from datasets constructed from different spectra on neural network training will be analyzed and discussed. In Figure 6 we have carried out three datasets, which are constructed over the spectra of the first region, second region, and full region, respectively. Therefore, the model accuracy trained with different datasets and noise addition with GridsearchCV and Bayesian Optimization are shown in Figure 9.

The results show that the accuracy of model trained by the first region dataset with GridsearchCV and Bayesian optimization is lower than the other region. It may be caused by the intervals between each normalized spectra curve at the spectral region are closer than the curves of the other regions. In the full region dataset, although it has the most complete data to obtain features, it is still affected by data of the first region, and it is difficult to achieve the best accuracy. In addition, since the data set constructed by the second region is highly distinguishable, the trained model shows excellent accuracy in every noise ratio.

Especially for more detailed analysis, we further explore the division of the spectral channel of the first and second region into four different sub-regions, respectively, in order to further analyze whether the neural network can obtain higher-precision measurements only through a few specific channels. Four sub-regions of first region spectra are as follows: the first sub-region denoted as S11 including three shorter wavelengths of 450 nm, 500 nm, and 550 nm; the second sub-region denoted as S12 including three middle wavelengths of 500 nm, 550 nm, and 570 nm; the third sub-region is denoted as S13 including three longer wavelengths of 570 nm, 600 nm, and 610 nm; and the last sub-region denoted as S14 including three wavelengths of 450 nm, 570 nm, and 610 nm over the first region. Furthermore, the second region spectra are also divided as follows: the first sub-region denoted as S21 including three shorter wavelengths of 650 nm, 680 nm, and 730 nm; the second sub-region denoted as S22 including three middle wavelengths of 730 nm, 760 nm, and 810 nm; the third sub-region is denoted as S23 including three longer wavelengths of 760 nm, 810 nm, and 860 nm; and the last sub-region is denoted as S24 including three wavelengths of 650 nm, 760 nm, and 860 nm over the second region. Figure 10 and Figure 11 shows the accuracy of the models trained by the above training sets of four sub-regions in each region under different noise ratios. It shows that the neural network trained by S11 and S23 which includes three longer wavelengths of 450 nm, 500 nm, 550 nm, and 760 nm, 810 nm, 860 nm performs the higher accuracy. It indicates that training dataset with more distinctive results can effectively improve the performance of the model. However, compared to the model trained by the six-channel of the second region, the model trained by S23 shows lower accuracy and it shows that the training data set still needs a sufficient number of wavelengths and high-quality data to effectively improve the performance of the model.

Although the spectral measurement with more features can improve the resolution ability of the neural network, the results reveal that the training with the dataset of the shorter six-channel of the first region is redundant. This analysis also shows that it is very important to construct an effective 1D-CNN model area for spectral measurement using the appropriate spectral region and number of wavelengths. This analysis shows that it is very important to construct an effective 1D-CNN model area for spectral measurement using the appropriate spectral region and number of wavelengths.

#### 4.2.2. Analysis of Data Augmentation with Noise Addition

In this section, the models trained for each condition of noise will be analyzed. Figure 9 shows that the model trained by noise-free addition has lower accuracy than the rest of the noise addition group. Therefore, by adding noise to the training dataset, the immunity of the neural network to noise, that is, the generalization ability of model will be greatly improved. In Figure 9, when the noise addition ratio increase, model accuracy is promoted. However, the accuracy declines slightly after 2%. Therefore, there are at least 2% of random noise is added to effectively improve the accuracy of the neural network. From Figure 9, the 2% noise addition will produce the greatest benefit to our optimized model. In summary, investigation of characteristics of the training data with proper noise will help to improve the anti-disturbances ability of neural network.

#### 4.2.3. Analysis of Model Optimization

According to the analysis in Section 4.1 and Section 4.2, this study chose the model trained by the 2nd region dataset with 2% noise addition. Moreover, Table 6 shows the most accuracy model hyperarameters obtained by BO, and Table 7 shows a more in-depth discussion and evaluation on the measurement ability of the model in SpO_2_ from 99% to 90%, which including the maximum error, average error, and standard deviation, so that the capability of measurement in each concentration can be realized clearly.

According to the analysis in Table 7, the maximum error of all concentrations is less than 2%, which is smaller than the maximum of traditional measurement deviation, proved that this new approach has higher degree of accuracy. Besides, the standard calibration is constructed by fixed condition of each SpO_2_ to fit the trend line of this CNN model and obtain the equation and R^2^ value. Figure 12 shows the calibration of predicted SpO_2_ for our proposed model according to the reading of SB100, which needs 8 s to reach stability, since the resolution of standard meter, Rossmax SB100, is 1% of the resolution of regular specification (which is far below the resolution of our proposed model as 0.1%). At the same time, another influencing factor worth considering is the measurement synchronization between the systems during the calibration measurement. Due to the time to achieve stability of output value in the measurement of different systems, and the physiologically stable distribution of SpO_2_, there will be also some influences which causes the less deviation between calibration and measurement. The linear regression analysis of proposed model that the maximum error of the model is smaller than the value of Table 7, and the slope of the fitting curve with linear form is close to 1. Therefore, it shows that the predicted result of the model is highly correlated with the measured value of SB100, and further strengthens the feasibility of this new approach. Furthermore, the R^2^ value is as high as 0.97, which shows that the predicted result of the model is highly correlated with the measured value of SB100, and further strengthens the feasibility of this new approach.

Moreover, in our previous study used DNN to measure SpO_2_, the total relative error is 0.76%, while the total relative error measuring by 1D-CNN model is 0.46%, which indicates that the accuracy of result has been highly improved. Besides, the results measuring by the 1D-CNN model have a lower standard deviation, which means that the overall measurement has a higher degree of stability.

### 4.3. Dynamic Measurement and Verification

To verify the validation of model and application of measurement system, a time-varying measurement of SpO_2_ is developed. In the following, the moving average filter will be applied to remove noise from the sensor reading. During the dynamic response measurement, the output of SpO_2_ concentration of spectral measurement based on our 1D-CNN model with optimization is compared with the reading of standard meter of SB100 at the same time. The SpO_2_ under test varies gradually from 98 to 82 for 135 samples within 94.5 s and the sampling time is 0.7 s for each reading.

Figure 13 shows three curves including the readings from the original prediction, SB100 and the signal after two filters. The green line represents the original data, and the orange line represents the signal after two filters: the median filter and the moving average filter. The signal of our proposed spectral measurement based on 1D-CNN Model is much more smooth and varies within a reasonable range compared to the reading of SB100, and even shows better signal quality than the readings of SB100.

## 5. Discussions and Conclusions

In this research, a technique for SpO_2_ of spectral measurement based on a 1D-CNN model is proposed and verified. In order to investigate the multiple spectral regions used for the deep learning of SpO_2_ measurement, we observed several spectral regions with large signal responses in SpO_2_ concentration and their boundary wavelengths to construct different spectral regions to analyze the validity of the proposed model. We found in the measurement in Figure 6 that the normalized spectral curve of the spectral region (450 nm, 500 nm, 550 nm, 570 nm, 600 nm, 610 nm) represented as the first region shows less response [13]. On the other hand, the normalized spectral curve of the second region (650 nm, 680 nm, 730 nm, 760 nm, 810 nm, 860 nm) is easier to distinguish since the spectral absorbance is relatively higher [13].

Three datasets for training and verification are built based on our analysis and it will also give the limitation of proposed model which can be possible improved by the hybrid deep neural network (HDNN) model with dropout of redundant spectral ranges. Our analysis shows it is extremely important to adopt an appropriate spectral region for building an effective 1D-CNN model region for the spectral measurement. The dataset is added with noise randomly, increasing the diversity of data and improving the generalization of the neural network. After that, the established dataset is delivered input to a 1D-CNN to obtain a measurement model of SpO_2_. Two optimization methods including GridSearchCV and Bayesian Optimization are applied to optimize the hyperparameters. The optimal accuracies of proposed model after optimization by GridSearchCV and Bayesian Optimization is 89.3% and 99.4%, respectively, trained with 2% random noise and the dataset at the spectral region of six wavelengths including 650 nm, 680 nm, 730 nm, 760 nm, 810 nm, and 860 nm. The total relative error of the best model is only 0.46%, optimized by Bayesian optimization. It shows that our proposed 1D-CNN model gives a new and feasible approach to measure SpO_2_ based on multi-wavelength. Moreover, we show that the multi-wavelength spectral measurement with deep learning architecture of 1D-CNN has low error in the elimination of the effect of tissue and has improved the accuracy of SpO_2,_ which can be widely applied to the other optical or radiometric spectroscopic measurement with complicated algorithms.

## Figures and Tables

**Figure 1 sensors-22-00328-f001:**
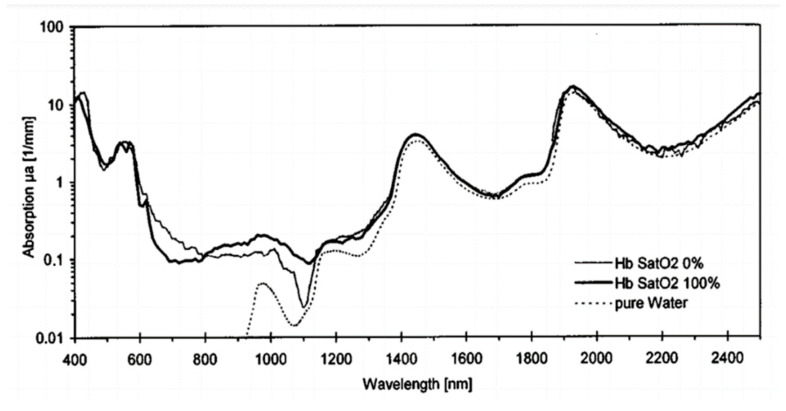
Different absorption of Hb and HbO_2_ in visible light and infrared light [16].

**Figure 2 sensors-22-00328-f002:**
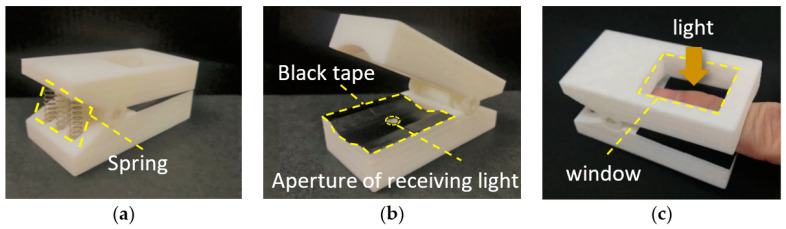
Design of 3D printed finger clip device (**a**) back side (**b**) front side (**c**) top side.

**Figure 3 sensors-22-00328-f003:**
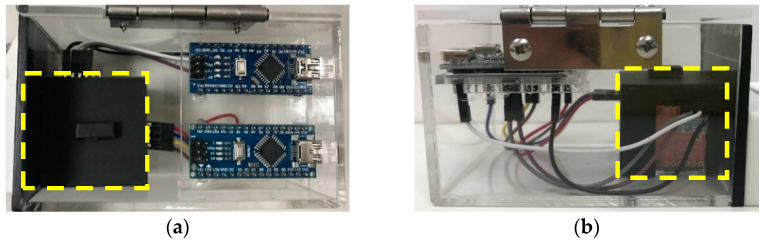
Design and assembly of 3D printed sensor light sensing case (**a**) top (**b**) back.

**Figure 4 sensors-22-00328-f004:**
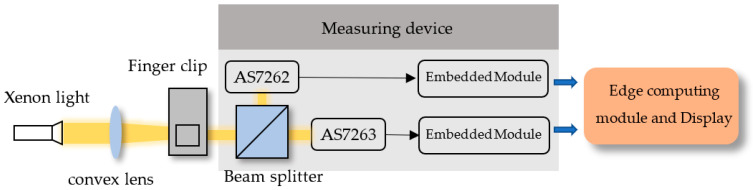
Architecture of light sensing and signal processing.

**Figure 5 sensors-22-00328-f005:**
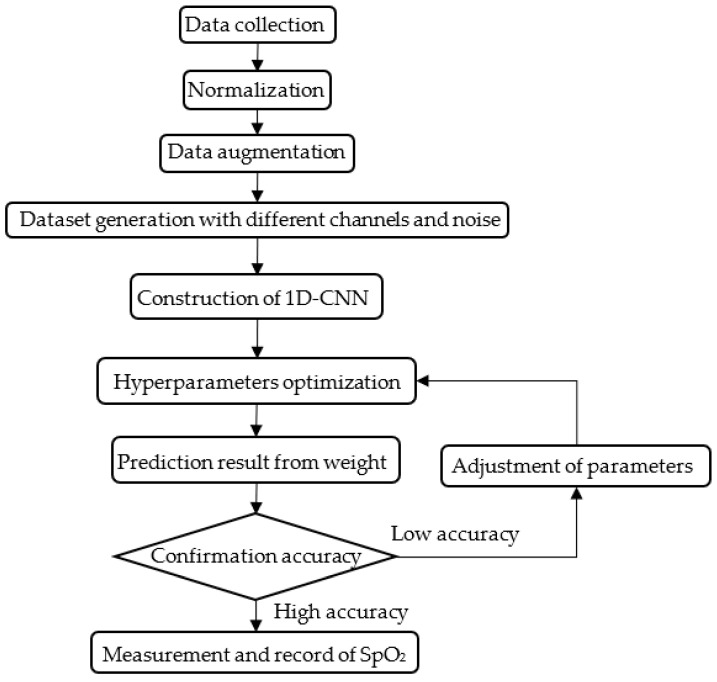
Experiment flow chart.

**Figure 6 sensors-22-00328-f006:**
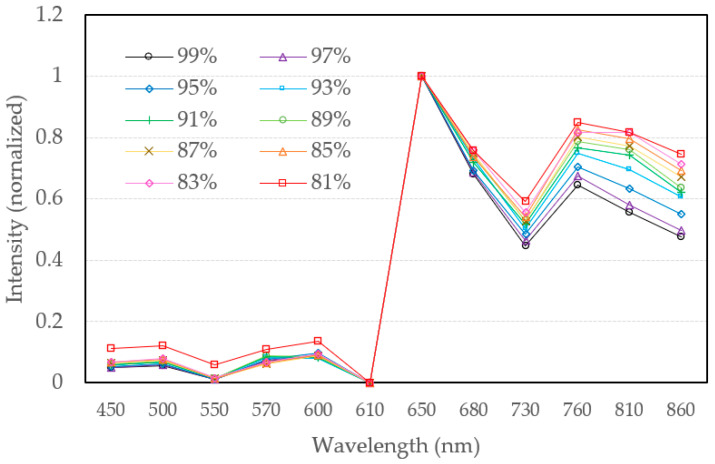
Normalized multi-channel spectral graphs with different concentrations.

**Figure 7 sensors-22-00328-f007:**
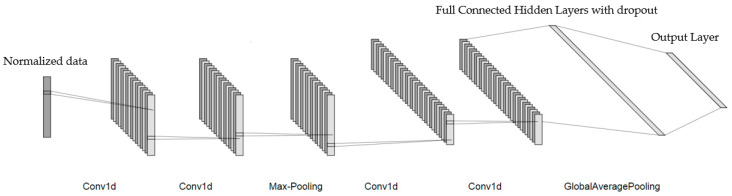
1D-CNN network architecture applied in this study.

**Figure 8 sensors-22-00328-f008:**
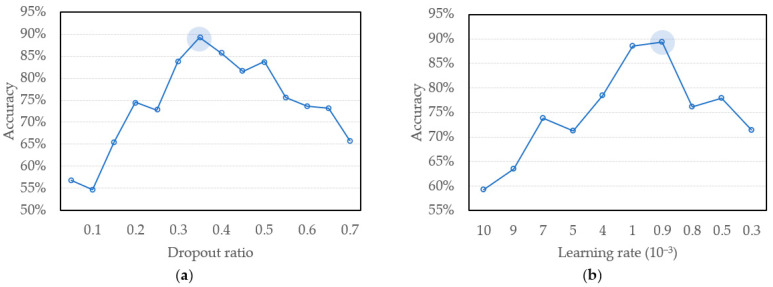
Accuracy of the best model in different hyperparameters from GridSearchCV (**a**) dropout ratio (**b**) learning rate.

**Figure 9 sensors-22-00328-f009:**
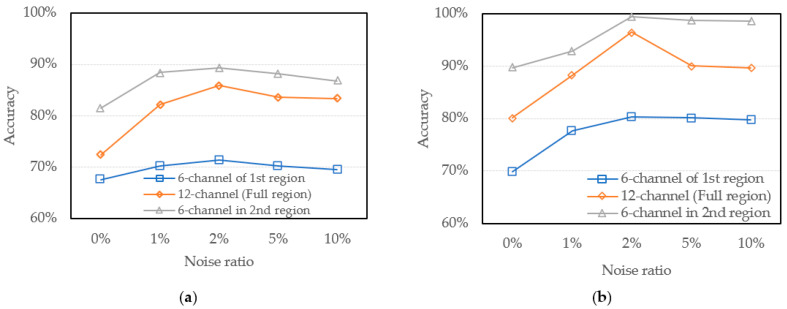
Model accuracy trained with different dataset and noise addition (**a**) GridsearchCV (**b**) Bayesian optimization.

**Figure 10 sensors-22-00328-f010:**
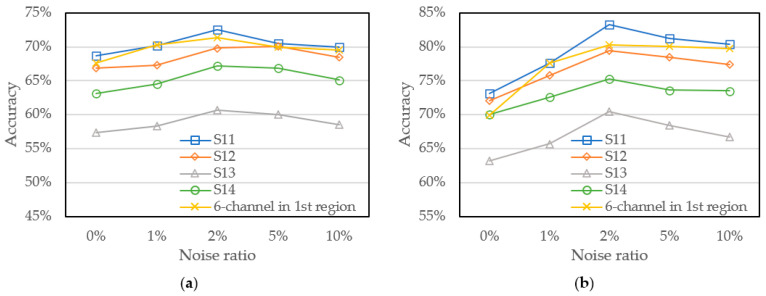
Model accuracy trained with divided 1st region and noise addition (**a**) GridsearchCV and (**b**) Bayesian optimization.

**Figure 11 sensors-22-00328-f011:**
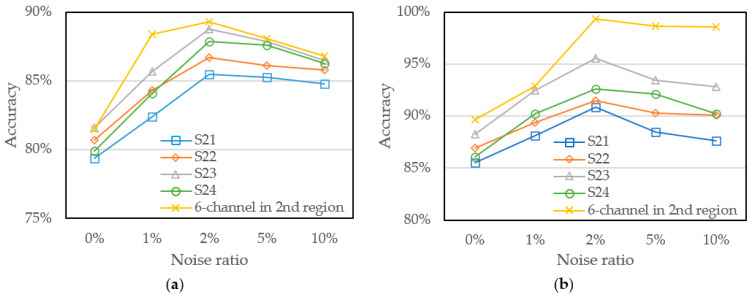
Model accuracy trained with divided second region and noise addition (**a**) GridsearchCV and (**b**) Bayesian optimization.

**Figure 12 sensors-22-00328-f012:**
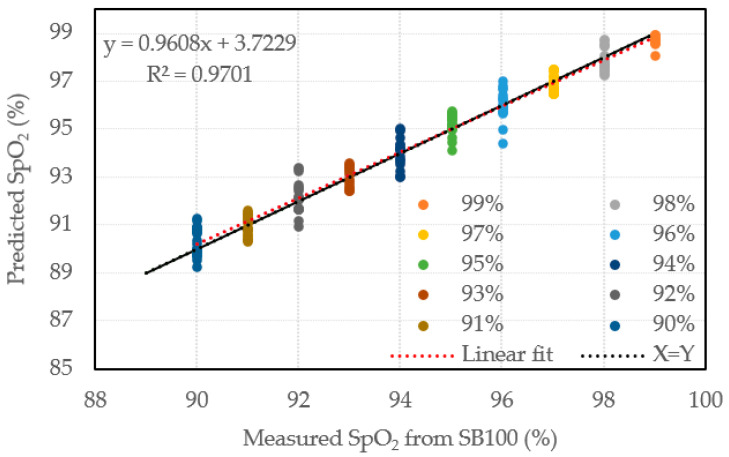
Standard calibration of SB100 and predicted SpO_2_ of proposed model.

**Figure 13 sensors-22-00328-f013:**
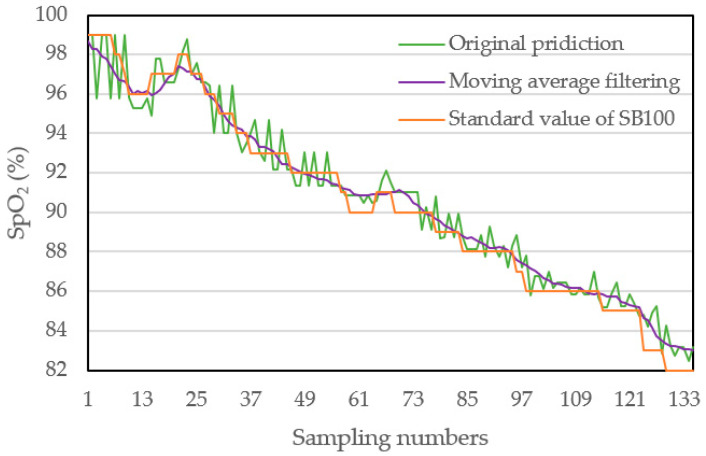
Original spectral data and measurement results after moving average filtering.

**Table 1 sensors-22-00328-t001:** Characteristic of AS7262, AS7263.

Characteristic	AS7262	AS7263	Unit
Sensor	Photodiode		[NA]
A/D Resolution	16		[bits]
Communication	UART or I^2^C		[NA]
Operating voltage	2.7–3.6		[V]
Temperature	−40 to 85		[°C]
FWHM	40	20	[nm]
Angle of incidence	±20		[°]
Integration time	2.8–714		[ms]
Channels	450, 500, 550, 570, 600, 650	610, 680, 730, 760, 810, 860	[nm]

**Table 2 sensors-22-00328-t002:** Searching conditions of hyperparameters for GridSearchCV.

Name	Range
Number of filters in 1st and 2nd CNN layers	2, 4, 8, 16, 32, 64
Number of filters in 3rd and 4th CNN layers	2, 4, 8, 16, 32, 64
Number of in full connect hidden layers	2, 4, 6, 8, 10
Kernel size	2, 4, 8, 16, 32
Dropout ratio	0.1, 0.2, …, 1
Number of nodes in full connect hidden layers	10, 20, …, 100
Learning rate	0.0001, 0.001, 0.01, 0.1
Batch size	10, 50, 100, 200, 300
epochs	10, 50, 100, 200, 300, 400, 500

**Table 3 sensors-22-00328-t003:** Searching conditions of hyperparameters for Bayesian optimization.

Name	Range
Number of filters in 1st and 2nd CNN layers	2 to 64
Number of filters in 3rd and 4th CNN layers	2 to 64
Number of in full connect hidden layers	1 to 10
Kernel size	2 to 32
Dropout ratio	0.01 to 1
Number of nodes in full connect hidden layers	1 to 100
Learning rate	0.0001 to 0.1
Batch size	10 to 300
epochs	10 to 500

**Table 4 sensors-22-00328-t004:** Accuracy of optimized model evaluated by testing dataset after GridSearchCV.

Dataset	Random Noise				
	0%	1%	2%	5%	10%
6-channel (1st region)	67.6%	70.3%	71.4%	70%	69.5%
12-channel (Full region)	72.4%	82.2%	85.9%	83.6%	83.4%
6-channel (2nd region)	81.5%	88.4%	89.3%	88.1%	86.8%

**Table 5 sensors-22-00328-t005:** Accuracy of optimized model evaluated by testing dataset after Bayesian optimization.

Dataset	Random Noise				
	0%	1%	2%	5%	10%
6-channel (1st region)	69.9%	77.6%	80.3%	80.1%	79.7%
12-channel (Full region)	80.1%	88.3%	96.4%	90%	89.6%
6-channel (2nd region)	89.7%	92.9%	99.4%	98.7%	98.6%

**Table 6 sensors-22-00328-t006:** Dataset, hyperarameters and noise ratio configuration of the best model.

Parameters	Value
Dataset	6-channel constructed by the 2nd region
Random noise ratio	2%
Number of filters in 1st and 2nd CNN layers	10
Number of filters in 3rd and 4th CNN layers	16
Number of full connect hidden layers	3
Kernel size	6
Dropout ratio	0.37
Number of nodes in full connect hidden layers	40
Learning rate	0.098
Batch size	85
epochs	100

**Table 7 sensors-22-00328-t007:** Deviation of each SpO_2_.

SpO_2_ (%)	Maximum Deviation (%)	Average Deviation ± Standard Deviation (%)
**99**	0.41	0.31 ± 0.18
**98**	0.77	0.46 ± 0.38
**97**	0.54	0.35 ± 0.27
**96**	1.08	0.45 ± 0.58
**95**	0.79	0.44 ± 0.36
**94**	1.14	0.56 ± 0.39
**93**	0.61	0.43 ± 0.22
**92**	1.51	0.58 ± 0.67
**91**	0.72	0.41 ± 0.34
**90**	1.43	0.61 ± 0.51
Total error: 0.46%		

## Data Availability

Data supporting reported results can be found from https://imeti.org/IMETI2021/download/Oral%20Session.pdf (accessed on 19 November 2021).

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
