# Peer review of "Research on Multiple Spectral Ranges with Deep Learning for SpO2 Measurement"

_sensors, 2022, doi:10.3390/s22010328_

Round 1
Reviewer 1 Report
This is fine
Author Response
The authors want to thank the unknown Reviewers and the Editor for their precious comments.
Reviewer 2 Report
I recommend this article for publication.
Author Response

(The authors gave the same response as above.)

Reviewer 3 Report
In this manuscript, the authors aimed to develop multiple spectral SpO2 measurement based on deep learning analysis (1D-CNN model). Below I list some comments for this manuscript.
The main concern is that the Discussion is too short. The authors can provide more description in these following topics.
- For the SpO2 measurement as described in p.6: the author can provide information for the accuracy of the present measurement. For Fig.12, the author can take this into account and discuss more about the performance of the present model.
- The authors can describe more about the advantage of using multiple wavelengths as features than using only two bands. Especially compare with Fig.6 (e.g., using only 860nm)
- The term “deep learning” is present in the title. The author can describe more about the advantage (especially in the discrimination performance) using DL than other methods.
- Conclusion: the author can suggest the meanings of the present findings for possible future application, and also the possible limitation of the present study.
Round 2
Reviewer 3 Report
The responses are satisfactory. I have no further comments.
This manuscript is a resubmission of an earlier submission. The following is a list of the peer review reports and author responses from that submission.
Round 1
Reviewer 1 Report
This paper describes a multi-wavelength optical device that can be used to measure SpO2. The authors use a neural network to process the data and demonstrate comparable performance to a commercial product. However, both the novelty and utility of the methods presented are unclear.
Major issues:
- The authors do not describe weaknesses associated with current 2 wavelength pulse oximeters. These devices are ubiquitous, inexpensive, and accurate enough for clinical use. There are limitations of this technique, but the authors do not present a compelling case for why multi-wavelength measurements are necessary
- The analysis method used by the authors relies on a 2 wavelength pulse oximeter to provide ground-truth training data. This means that the proposed device can never exceed the performance of the 2 wavelength version
- The authors make many statements of dubious accuracy without providing citations. One example, "When multi wavelength is used there is currently no general formula that can be used." The modified Beer-Lambert law is one possible formula. Regardless, statements such as this should be supported by references in the literature.
- There is not enough information to evaluate the data collection process. How many measurements were collected? From how many subjects? etc.
- Studies involving human subjects should be approved by an ethics board and a statement made to that effect.
Reviewer 2 Report
The scientific interest of the paper is good and is in tune with the times. However, this document may be improved for publication in an international journal. In particular, bibliography must be more extensive, the contribution of the paper compared to the existing must be more highlighted at the end of the introduction and the interpretation and comments on the results must be deepened. In addition, for scientific progress, it is advisable to make the sources available to the reader on a Github-type sharing platform.
In the attached document, some remarks are made.

Round 2
Reviewer 1 Report
I still do not feel that this paper is novel or interesting enough for publication. While the authors responded to some of my critiques in a cursory way, they failed to address the deeper, more serious concerns I raised.
Author Response
Please see the attachment。
